# LMSeg: Language-guided Multi-dataset Segmentation

**Qiang Zhou[1], Yuang Liu[2], Chaohui Yu[1], JingLiang Li[3], Zhibin Wang[1], Fan Wang[1]***
[1]Alibaba Group  [2]East China Normal University
[3]University of the Chinese Academy of Sciences
{jianchong.zq,huakun.ych,zhibin.waz,fan.w}@alibaba-inc.com
frankliu624@gmail.com, lijingliang20@mails.ucas.ac.cn

## Abstract

It's a meaningful and attractive topic to build a general and inclusive segmentation model that can recognize more categories in various scenarios. A straightforward way is to combine the existing fragmented segmentation datasets and train a multi-dataset network. However, there are two major issues with multi-dataset segmentation: (i) the inconsistent taxonomy demands manual reconciliation to construct a unified taxonomy; (ii) the inflexible one-hot common taxonomy causes time-consuming model retraining and defective supervision of unlabeled categories. In this paper, we investigate the multi-dataset segmentation and propose a scalable Language-guided Multi-dataset Segmentation framework, dubbed LMSeg, which supports both semantic and panoptic segmentation. Specifically, we introduce a pre-trained text encoder to map the category names to a text embedding space as a unified taxonomy, instead of using inflexible one-hot label. The model dynamically aligns the segment queries with the category embeddings. Instead of relabeling each dataset with the unified taxonomy, a category-guided decoding module is designed to dynamically guide predictions to each dataset's taxonomy. Furthermore, we adopt a dataset-aware augmentation strategy that assigns each dataset a specific image augmentation pipeline, which can suit the properties of images from different datasets. Extensive experiments demonstrate that our method achieves significant improvements on four semantic and three panoptic segmentation datasets, and the ablation study evaluates the effectiveness of each component.

## 1 Introduction

Image Segmentation has been a longstanding challenge in computer vision and plays a pivotal role in a wide variety of applications ranging from autonomous driving (Levinson et al., 2011; Maurer et al., 2016) to remote sensing image analysis (Ghassemian, 2016). Building a general and inclusive segmentation model is meaningful to real-world applications. However, due to the limitation of data collection and annotation cost, there are only fragmented segmentation datasets of various scenarios available, such as ADE20K (Zhou et al., 2017), Cityscapes (Cordts et al., 2016), COCO-stuff (Caesar et al., 2018), *etc*. Meanwhile, most work of segmentation (Long et al., 2015; Chen et al., 2018; Zheng et al., 2021) focus on single-dataset case, and overlook the generalization of the deep neural networks. Generally, for different data scenarios, a new set of network weights are supposed to be trained. As a compromise of expensive images and annotations for all scenarios, how to construct a multi-dataset segmentation model with the existing fragmented datasets is attractive for supporting more scenarios.

The primary issue of multi-dataset learning is the inconsistent taxonomy, including category coincidence, ID conflict, naming differences, *etc*. For example, the category of "person" in ADE20k dataset are labeled as "person" and "rider" in Cityscapes dataset. As shown in Figure 1(a), Lambert et al. (2020) manually establish a unified taxonomy with the one-hot label, relabel each dataset, and then train a segmentation model for all involved datasets, which is time-consuming and error-prone.

---

*Corresponding author.

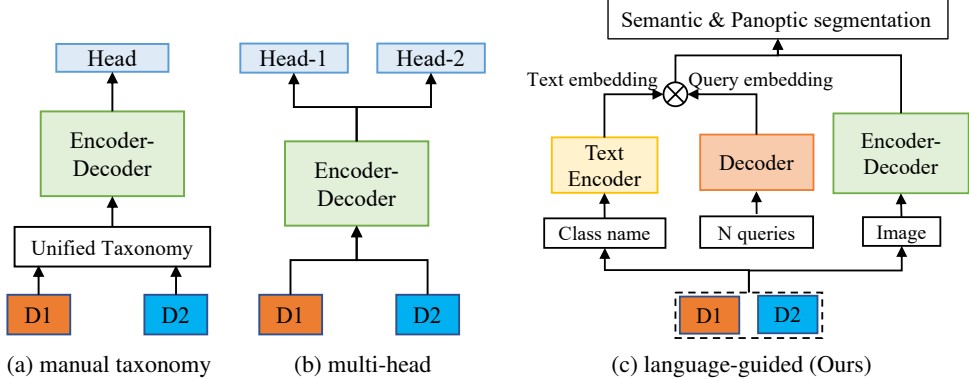

Figure 1: Comparison of different multi-dataset segmentation approaches.

Moreover, the one-hot taxonomy is inflexible and unscalable. When extending the datasets or categories, the unified taxonomy demands reconstruction and the model requires retraining. A group of advanced researches (Wang et al., 2022) utilizes multi-head architecture to train a weight-shared encoder-decoder module and multiple dataset-specific headers, as shown in Figure 1(b). The multi-head approach is a simple extension of traditional single-dataset learning, not convenient during inference. For example, to choose the appropriate segmentation head, which dataset the test image comes from needs to be predefined or specified during inference.

To cope with these challenges, we propose a language-guided multi-dataset segmentation (LMSeg) framework that supports both semantic and panoptic segmentation tasks (Figure 1(c)). On the one hand, in contrast to manual one-hot taxonomy, we introduce a pre-trained text encoder to automatically map the category identification to a unified representation, *i.e.*, text embedding space. The image encoder extracts pixel-level features, while the query decoder bridges the text and image encoder and associates the text embeddings with the segment queries. Figure 2 depicts the core of text-driven taxonomy for segmentation. As we can see that the text embeddings of categories reflect the semantic relationship among the classes, which cannot be expressed by one-hot labels. Thus, the text-driven taxonomy can be extended infinitely without any manual reconstruction. On the other hand, instead of relabeling each dataset with a unified taxonomy, we dynamically redirect the model's predictions to each dataset's taxonomy. To this end, we introduce a category-guided decoding (CGD) module to guide the model to predict involved labels for the specified taxonomy. In addition, the image properties of different datasets are various, such as resolution, style, ratio, *etc*. And, applying appropriate data augmentation strategy is necessary. Therefore, we design a dataset-aware augmentation (DAA) strategy to cope with this. In a nutshell, our contributions are four-fold:

- We propose a novel approach for multi-dataset semantic and panoptic segmentation, using text-query alignment to address the issue of taxonomy inconsistency.
- To bridge the gap between cross-dataset predictions and per-dataset annotations, we design a category-guided decoding module to dynamically guide predictions to each dataset's taxonomy.
- A dataset-aware augmentation strategy is introduced to adapt the optimal preprocessing pipeline for different dataset properties.
- The proposed method achieves significant improvements on four semantic and three panoptic segmentation datasets.

## 2 RELATED WORK

### 2.1 SEMANTIC SEGMENTATION

As a dense prediction task, semantic segmentation plays a key role in high-level scene understanding. Since the pioneering work of fully convolutional networks (FCNs) (Long et al., 2015), pixel-

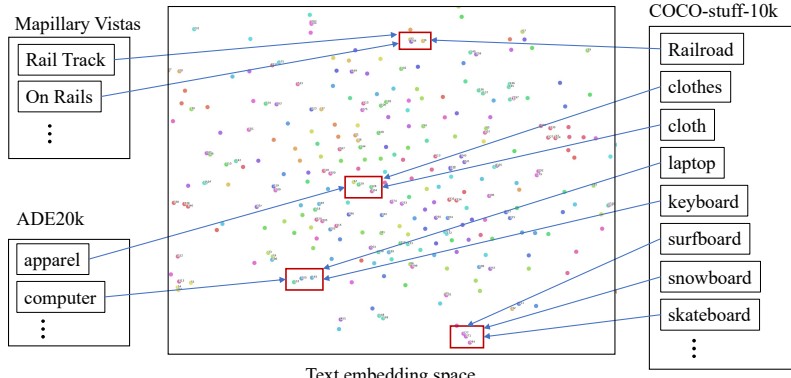

Figure 2: t-SNE (van der Maaten & Hinton, 2008) visualization of category embeddings for several semantic segmentation datasets using CLIP's text encoder for feature extraction. The text embedding space of CLIP is suitable as a unified taxonomy, with semantically similar categories holding closer text embeddings.

wise classification has become the dominant approach for deep learning-based semantic segmentation. After FCNs, most semantic segmentation models focus on aggregating long-range context in the final feature map. These methods include atrous convolutions with different atrous rates (Chen et al., 2018; 2017), pooling operations with varying kernel sizes (Zhao et al., 2017), and variants of non-local blocks (Fu et al., 2019; Yuan et al., 2021; Huang et al., 2019; Wang et al., 2018). More recently, SETR (Zheng et al., 2021) and Segmenter (Strudel et al., 2021) replace the traditional convolutional backbone with Vision Transformers (ViT) (Dosovitskiy et al., 2021) to capture long-range context from the first layer. Pixel-wise semantic segmentation methods are difficult to extend to instance-level segmentation. Some mask classification-based methods (Carion et al., 2020; Wang et al., 2021) have recently emerged to unify semantic and instance segmentation tasks. The most representative work is MaskFormer (Cheng et al., 2021; 2022), which solves these two segmentation tasks in a unified manner. However, these segmentation methods typically follow the setting of single-dataset training and can not maintain high accuracy on other datasets without finetuning.

## 2.2 MULTI-DATASEST TRAINING

Compared to single-dataset learning, recently multi-dataset learning has received increasing attention in consequence of its robustness and generalization. Perrett & Damen (2017) apply multi-dataset learning to action recognition during the pre-training stage. To settle the challenge of multi-dataset object detection, Zhao et al. (2020) propose a pseudo-labeling strategy and Yao et al. (2020) propose a dataset-aware classification loss. To avoid manually building unified taxonomy, Zhou et al. (2022c) propose a formulation to automatically integrate the dataset-specific outputs of the partitioned detector into a unified semantic taxonomy. For multi-dataset semantic segmentation, MSeg (Lambert et al., 2020) manually creates a common taxonomy to unite segmentation datasets from multiple domains. Shi et al. (2021) first pre-trains the network on multiple datasets with a contrast loss and then fine-tunes it on specific datasets. Based on the multi-head architecture, CDCL (Wang et al., 2022) proposes a dataset-aware block to capture the heterogeneous statistics of different datasets. Yin et al. (2022) utilize text embedding to improve the zero-shot performance on semantic segmentation and claim to adopt sentences instead of category names for better text embedding. In contrast to Yin et al. (2022), we focus on more general multi-dataset segmentation, including semantic and panoptic segmentation, and further propose solutions for the problem of incomplete annotation in multi-dataset training.

## 2.3 VISION-LANGUAGE PRE-TRAINING

Vision-Language Pre-training (VLP) has achieved significant progress in the last few years, which aims to jointly obtain a pair of image and text encoders that can be applied to numerous multi-modal tasks. CLIP (Radford et al., 2021) is a landmark work of VLP, which jointly trains the encoders on 400 million image-text pairs collected from the web. Following CLIP, many researchers attempt

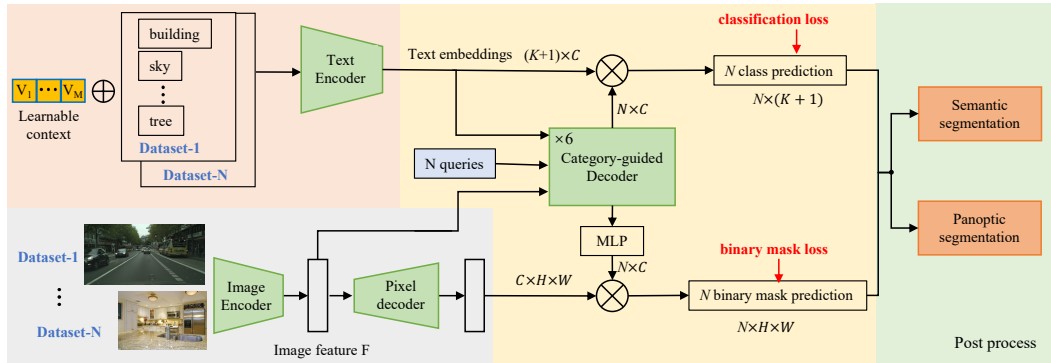

Figure 3: LMSeg: language-guided multi-dataset segmentation framework.

to transfer the CLIP model to downstream tasks. Zhou et al. (2022b;a); Gao et al. (2021); Zhang et al. (2021) show that CLIP can help to achieve state-of-the-art performance on few-shot or even zero-shot classification tasks. DenseCLIP (Rao et al., 2022) applies CLIP to dense prediction tasks via language-guided fine-tuning and achieves promising results. We claim that the text embedding space of CLIP can be regarded as a unified taxonomy, as shown in Figure 2, and propose text-query alignment instead of a softmax classifier for multi-dataset segmentation.

## 3 METHOD

### 3.1 OVERVIEW

In this section, we introduce the proposed LMSeg framework, a new language-guided multi-dataset segmentation framework, supporting both semantic and panoptic segmentation under multi-dataset learning. As shown in Figure 3, the inputs consist of an image and a set of class names corresponding to the dataset to which the image belongs. The LMSeg is decomposed of an encoder-decoder pixel feature extractor, a pre-trained text encoder, a Transformer decoder with category-guided module and a dataset-aware augmentation strategy. The image is first preprocessed using the proposed dataset-aware augmentation strategy, and then image features are extracted through the image encoder and pixel decoder. The class names are mapped to text embeddings by the pre-trained text encoder. The category-guided decoding module bridges the text embeddings and the image features to semantic and panoptic mask predictions. We detail each component of our framework in the following.

### 3.2 TEXT ENCODER AND PIXEL FEATURE EXTRACTOR

The text encoder takes the class names of a dataset as input and outputs the corresponding text embeddings $\epsilon_{\text{text}}^k \in \mathbb{R}^{(K_k+1) \times C_t}$ for that dataset, where $K_k$ is the number of classes of the $k$-th dataset, and $C_t$ is the text embedding dimension. The implementation of the text encoder follows CLIP (Radford et al., 2021). During the training phase, the parameters of the text encoder are initialized with a pre-trained CLIP model and fixed without parameter update. The text encoder's output dimension $C_t$ is incompatible with the segment query embedding, and we use a linear adapter layer to adjust the dimension of the text embeddings. It is worth noting that we can store the text embeddings of class names after training. That is, by reducing the overhead introduced by the text encoder, the inference time of LMSeg is barely increased compared to other segmentation frameworks (Cheng et al., 2021). Existing work show that, text prompt (e.g., "a photo of [class name]") can improve the transfer performance of pre-trained CLIP models. Inspired by CoOp (Zhou et al., 2022b), when generating the text embedings $\epsilon_{\text{text}}^k$, we utilize a learnable prompt, namely "[v]$_1$[v]$_2$...[v]$_L$[class name]", where $L$ is the length of the learnable vectors and is set to 8 by default. Note that the learnable vectors are shared across all datasets.

The pixel feature extractor takes an image of size $3 \times H \times W$ as input. The image encoder first converts the image into a low-resolution image feature map $\mathcal{F} \in \mathbb{R}^{C_{\mathcal{F}} \times H' \times W'}$, and then the pixel decoder gradually upsamples the feature map to generate per-pixel embeddings $\epsilon_{\text{pixel}} \in \mathbb{R}^{C_\epsilon \times H \times W}$,

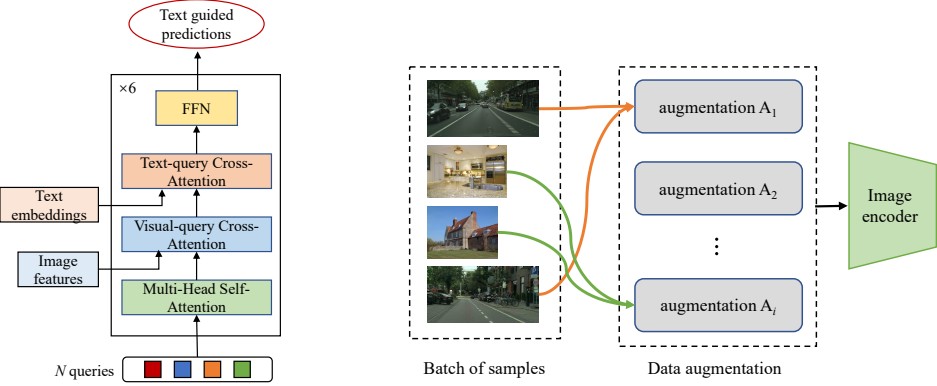

Figure 4: Category-guided decoding (CGD).    Figure 5: Dataset-aware augmentation (DAA).

where $C_\epsilon$ is the embedding dimension. Finally, we obtain each binary mask prediction $m_i \in [0,1]^{H \times W}$ via a dot product between the $i$-th segment query embedding and the per-pixel embeddings $\epsilon_{\text{pixel}}$. The image encoder can use arbitrary backbone models, not limited to the ResNet (He et al., 2016) as we use in this work.

### 3.3 TEXT-QUERY ALIGNMENT

In multi-dataset segmentation, establishing a unified taxonomy is often an essential step, which is usually time-consuming and error-prone, especially as the number of categories increases. In this subsection, we introduce the proposed text-query alignment, which uses the text embedding as a unified taxonomy representation without manually construction.

Specifically, we remove the static classifier layer (usually a fully-connected layer), which transforms the segment query embeddings into the final classification logits. Instead, we align the segment query embeddings $\epsilon_{\text{query}} \in \mathbb{R}^{N \times C}$ and the text embeddings $\epsilon_{\text{text}} \in \mathbb{R}^{(K+1) \times C}$ via a contrastive loss. Taking the samples of $k$-th dataset as an example, we dot product the text embeddings and query embeddings to predict the classification logits $\mathbf{p}^k \in \mathbb{R}^{N \times (K_k+1)}$ for $N$ segment queries:

$$\mathbf{p}^k = \frac{\epsilon_{\text{query}} \cdot \left(\epsilon_{\text{text}}^k\right)^T}{\tau}, \tag{1}$$

where $\tau$ is the temperature parameter and is set to 0.07 by default. The $\epsilon_{\text{query}} \in \mathbb{R}^{N \times C}$ denotes the $N$ query embeddings with dimension $C$, which is the output of the category-guided decoding (CGD) module in Figure 3. The $\epsilon_{\text{text}}^k \in \mathbb{R}^{(K_k+1) \times C}$ represents the text embedding for the $K_k + 1$ categories of the $k$-th dataset (including one "background" category), and is the output of the text encoder module in Figure 3. The dynamic classification logits $\mathbf{p}^k$ are supervised with a contrastive objectives $\mathcal{L}_{\text{cl}}$ as:

$$\mathcal{L}_{\text{cl}} = \frac{1}{N} \sum_{i=0}^{N-1} \left( -\log \frac{\exp(\mathbf{p}_{i,j+}^k)}{\sum_{j=0}^{K_k} \exp(\mathbf{p}_{i,j}^k)} \right), \tag{2}$$

where the outer sum is over $N$ segment queries. For the $i$-th query, the loss is the log loss of a $(K_k + 1)$-way softmax-based classifier that tries to classify the $i$-th query as $j+$.

### 3.4 CATEGORY-GUIDED DECODING MODULE

The text-query alignment does not entirely address the issue of taxonomy differences in multi-dataset segmentation. For example, the query embedding corresponding to a human might be forced to align with the text embedding of "person" or "rider" in the Cityscapes dataset but only to "person" in the ADE20K dataset. That is, for input images, the alignment targets of query embeddings are non-deterministic (varies with the dataset), which affects the performance of multi-dataset segmentation, as our experiments demonstrate. Instead of relabeling each dataset with a unified taxonomy, we propose a simpler approach that dynamically redirects the model's predictions to each dataset's taxonomy. Through prediction redirection, we can arbitrarily specify the categories that the model needs to predict so that we can use the original annotations of each dataset.

| Total training cost | Backbone | Model | Pre-train | M.D.T | Per-dataset mIoU | | | | Average mIoU |
|---|---|---|---|---|---|---|---|---|---|
| | | | | | ADE20K | Cityscapes | COCO-Stuff | Mapillary Vistas | |
| 160k | R50 | MaskFormer | ImageNet | | 40.24 | 75.43 | 36.83 | 46.72 | 49.80 |
| | | MaskFormer | CLIP | | 39.75 | 77.30 | 38.50 | 47.37 | 50.73 |
| | | LMSeg (Ours) | CLIP | ✓ | 41.37 | 79.13 | 39.01 | 50.82 | **52.58** |
| 320k | R50 | MaskFormer | ImageNet | | 42.55 | 76.56 | 36.33 | 50.45 | 51.47 |
| | | MaskFormer | CLIP | | 43.18 | 78.20 | 38.69 | 52.15 | 53.05 |
| | | LMSeg (Ours) | CLIP | ✓ | 44.89 | 79.81 | 39.21 | 52.27 | **54.04** |
| 640k | R50 | MaskFormer | ImageNet | | 44.57 | 76.37 | 35.89 | 53.70 | 52.63 |
| | | MaskFormer | CLIP | | 45.66 | 77.95 | 38.16 | 53.16 | 53.73 |
| | | LMSeg (Ours) | CLIP | ✓ | 45.16 | 80.93 | 38.60 | 54.34 | **54.75** |

Table 1: Semantic segmentation accuracy (mIoU) compared with single-dataset training models. M.D.T denotes multi-dataset training.

We propose a category-guided decoding module to dynamically adapt to classes to be predicted by the model, as shown in Figure 4. The decoder module follows the standard architecture of the transformer, using multi-head self- and cross-attention mechanisms and an FFN module to transform $N$ segment queries $\epsilon_{\text{query}}$. The self-attention to query embeddings enables the model to make global inferences for all masks using pairwise relationships between them. The cross-attention between query embeddings and image features $\mathcal{F}$ is able to use the entire image as context. Finally, the cross-attention between query embeddings $\epsilon_{\text{query}}$ and text embeddings $\epsilon_{\text{text}}$ guides the query embeddings to the classes corresponding to the input text embeddings.

We use 6 decoder modules with 100 segment queries by default. The $N$ segment queries are initialized as zero vectors, each associated with a learnable positional encoding.

## 3.5 DATASET-AWARE AUGMENTATION

Different datasets usually have different characteristics (*e.g.*, image resolution) and therefore different data augmentations are used for training in general. For example, in MaskFormer (Cheng et al., 2021), a crop size of $512 \times 1024$ is used for Cityscapes, while a crop size of $512 \times 512$ for ADE20K. For multi-dataset training, MSeg (Lambert et al., 2020) first upsamples the low-resolution training images with a super-resolution model, then uses the same data augmentation to train all datasets. Unlike MSeg, we propose a dataset-aware augmentation strategy, shown in Figure 5, which is simpler and more scalable. Specifically, for each training sample, we determine which dataset the sample comes from and choose the corresponding augmentation strategy $A_i$. Furthermore, the dataset-aware augmentation strategy allows us to make fair comparisons with single-dataset training models while keeping the data augmentation for each dataset the same.

## 3.6 TOTAL TRAINING OBJECTIVE

The total training objective of LMSeg consists of a contrastive loss $\mathcal{L}_{\text{cl}}$ and two binary mask losses, focal loss $\mathcal{L}_{\text{focal}}$ (Lin et al., 2017) and dice loss $\mathcal{L}_{\text{dice}}$ (Milletari et al., 2016), which are applied to the batch of samples from multiple datasets:

$$\mathcal{L} = \sum_{k=1}^{M} \sum_{n=1}^{N^k} \left( \mathcal{L}_{\text{cl}} + \lambda_{\text{focal}} \cdot \mathcal{L}_{\text{focal}} + \lambda_{\text{dice}} \cdot \mathcal{L}_{\text{dice}} \right) , \tag{3}$$

where $M$ is the number of datasets. $N^k$ is the number of training samples from the $k$-th dataset. The hyper-parameters $\lambda_{\text{focal}}$ and $\lambda_{\text{dice}}$ are set to 20.0 and 1.0 by default. The weight for the "no object" ($\emptyset$) in the contrastive loss $\mathcal{L}_{\text{cl}}$ is set to 0.1.

Before computing the loss function, we have to figure out how to assign the ground truth targets to the model's predictions since LMSeg outputs $N$ unordered predictions. We take one image from the $k$-th dataset as an example to illustrate how the ground truth is assigned during training. We denote $\bar{y} = \{(\bar{c}_i^k, \bar{m}_i^k) | i = 1, \cdots, \bar{N}\}$ as the ground truth set of masks in the image, where $\bar{c}_i^k$ is the label and $\bar{m}_i^k$ represents the binary mask. LMSeg infers a fixed-size set of $N$ unordered predictions $y = \{(p_i^k, m_i^k) | i = 1, \cdots, N\}$ for the input image. Assuming that $N$ is larger than the number of target masks in the image, we pad set $\bar{y}$ with $\emptyset$ (no object) to the size of $N$. To find a bipartite matching between these two sets we search for a permutation of $N$ elements $\sigma \in \mathfrak{S}_N$ with the lowest cost:

$$\hat{\sigma} = \arg\min_{\sigma \in \mathfrak{S}_N} \sum_{i}^{N} \mathcal{L}_{\text{match}}(y_i, \bar{y}_{\sigma(i)}) , \tag{4}$$

| Total training cost | Backbone | Model | Pre-train | Per-dataset mIoU | | | | Average mIoU |
|---|---|---|---|---|---|---|---|---|
| | | | | ADE20K | Cityscapes | COCO-Stuff | Mapillary Vistas | |
| 160k | R50 | Manual-taxonomy | CLIP | 40.24 | 79.37 | 38.36 | 46.70 | 51.16 |
| | | Multi-head | CLIP | 41.19 | 79.29 | 38.26 | 49.25 | 51.99 |
| | | LMSeg (Ours) | CLIP | 41.37 | 79.13 | 39.01 | 50.82 | **52.58** |
| 320k | R50 | Manual-taxonomy | CLIP | 43.46 | 80.23 | 39.01 | 51.58 | 53.56 |
| | | Multi-head | CLIP | 44.30 | 80.06 | 38.57 | 52.13 | 53.76 |
| | | LMSeg (Ours) | CLIP | 44.89 | 79.81 | 39.21 | 52.27 | **54.04** |
| 640k | R50 | Manual-taxonomy | CLIP | 45.06 | 80.71 | 39.43 | 52.95 | 54.53 |
| | | Multi-head | CLIP | 46.47 | 80.07 | 37.73 | 53.22 | 54.37 |
| | | LMSeg (Ours) | CLIP | 45.16 | 80.93 | 38.60 | 54.34 | **54.75** |

Table 2: Semantic segmentation accuracy (mIoU) compared with multi-dataset models.

| Training Setting | Backbone | Method | Pre-train | Per-dataset PQ (number of classes) | | | Average PQ |
|---|---|---|---|---|---|---|---|
| | | | | ADE20K-Panoptic | Cityscapes-Panoptic | COCO-Panoptic | |
| 320k | R50 | Manual-taxonomy | CLIP | 25.88 | 51.50 | 31.26 | 36.21 |
| | | Multi-head | CLIP | 23.51 | 48.56 | 27.61 | 33.22 |
| | | LMSeg (Ours) | CLIP | 30.84 | 51.75 | 34.16 | **38.91** |
| 640k | R50 | Manual-taxonomy | CLIP | 28.65 | 52.18 | 34.76 | 38.53 |
| | | multi-head | CLIP | 28.66 | 50.05 | 31.27 | 36.66 |
| | | LMSeg (Ours) | CLIP | 34.20 | 55.28 | 37.72 | **42.40** |
| 960k | R50 | Manual-taxonomy | CLIP | 30.68 | 53.47 | 36.63 | 40.26 |
| | | multi-head | CLIP | 31.99 | 54.17 | 35.18 | 40.44 |
| | | LMSeg (Ours) | CLIP | 35.43 | 54.77 | 38.55 | **42.91** |

Table 3: Panoptic segmentation accuracy (PQ) compared with other multi-dataset training methods.

where the matching cost $\mathcal{L}_{\text{match}}(y_i, \bar{y}_{\sigma(i)})$ takes into account the class prediction and the similarity of predicted and ground truth masks,

$$\mathcal{L}_{\text{match}}(y_i, \bar{y}_{\sigma(i)}) = \begin{cases} -p_i^k(\bar{c}_{\sigma(i)}^k) + \lambda_{\text{focal}} \cdot \mathcal{L}_{\text{focal}}(m_i^k, \bar{m}_{\sigma(i)}^k) + \lambda_{\text{dice}} \cdot \mathcal{L}_{\text{dice}}(m_i^k, \bar{m}_{\sigma(i)}^k), & \bar{c}_{\sigma(i)}^k \neq \varnothing \\ +\infty, & \bar{c}_{\sigma(i)}^k = \varnothing \end{cases} \quad (5)$$

During the training phase, we compute the optimal assignment $\hat{\sigma}$ for each sample in the batch and then accumulate the loss function of Equation 3 over these samples.

## 4 EXPERIMENTS

### 4.1 IMPLEMENTATION DETAILS

**Datasets.** For semantic segmentation, we evaluate on four public semantic segmentation datasets: ADE20K (Zhou et al., 2017) (150 classes, containing 20k images for training and 2k images for validation), COCO-Stuff-10K (Caesar et al., 2018) (171 classes, containing 9k images for training and 1k images for testing), Cityscapes (Cordts et al., 2016) (19 classes, containing 2975 images for training, 500 images for validation and 1525 images for testing), and Mapillary Vistas (Neuhold et al., 2017) (65 classes, containing 18k images for training, 2k images for validation and 5k images for testing). For panoptic segmentation, we use COCO-Panoptic (Lin et al., 2014) (80 "things" and 53 "stuff" categories), ADE20K-Panoptic (Zhou et al., 2017) (100 "things" and 50 "stuff" categories) and Cityscapes-Panoptic (Cordts et al., 2016) (8 "things" and 11 "stuff" categories).

**Training setup.** We use Detectron2 (Wu et al., 2019) to implement our LMSeg. Without specific instruction, dataset-aware augmentation is adopted and the same as MaskFormer (Cheng et al., 2021) for each dataset. We use AdamW (Loshchilov & Hutter, 2019) and the poly (Chen et al., 2018) learning rate schedule with an initial learning rate of $1e^{-4}$ and a weight decay of $1e^{-4}$. Image encoders are initialized with pre-trained CLIP (Radford et al., 2021) weights. Following MaskFormer, a learning rate multiplier of 0.1 is applied to image encoders. Other common data augmentation strategies like random scale jittering, random horizontal flipping, random cropping and random color jittering are utilized. For the ADE20K dataset, we use a crop size of $512 \times 512$. For the Cityscapes dataset, we use a crop size of $512 \times 1024$. For the COCO-Stuff-10k dataset, we use a crop size of $640 \times 640$. For the Mapillary Vistas dataset, we use a crop size of $1280 \times 1280$. All models are trained with 8 A100 GPUs and a batch size of 16. Each image in the batch is randomly sampled from all datasets. For panoptic segmentation, we follow exactly the same architecture, loss, and training procedure as

we use for semantic segmentation. The only difference is supervision: *i.e.*, category region masks in semantic segmentation vs. object instance masks in panoptic segmentation. The data augmentation of each dataset follows MaskFormer, and we also provide the detail of augmentation in the appendix.

## 4.2 RESULTS AND COMPARISONS

When conducting multi-dataset training, we train a single LMSeg model and report the model's performance on each dataset. While conducting single-dataset training, we train a separate MaskFormer model for each dataset, and the total training cost is cumulative over all datasets.

When comparing with multi-dataset training methods, we re-implement the "multi-head" and "manual-taxonomy" methods based on MaskFormer for a fair comparison. For the multi-head method denoted by "MaskFormer + Multi-head" in Table 2 and Table 3, we share the components of MaskFormer on various datasets, except for the classification head. For the manual-taxonomy method denoted by "MaskFormer + Manual-taxonomy" in Table 2 and Table 3, we manually construct the unified taxonomy. For simplicity, we only unite duplicate class names across datasets. Semantically similar classes such as "apparel" and "clothes" are not split or merged.

### 4.2.1 MULTI-DATASET SEMANTIC SEGMENTATION

Table 1 depicts the comparison between LMSeg and single-dataset training models. Our LMSeg outperforms single-dataset training models in all settings. This experimental results show that compared with single-dataset training, multi-dataset training can obtain a single robust model capable of recognizing more categories and improving the average performance on multiple datasets. For a more intuitive comparison, we plot the average mIoU w.r.t. total training cost for four datasets in Figure 6.

Table 2 shows the comparison between LMSeg and other multi-dataset training methods. Experimental results show that LMSeg outperforms "multi-head" and "manual-taxonomy" under various settings.

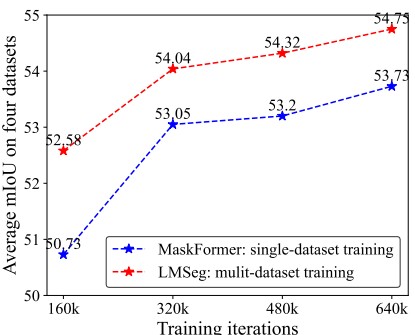

Figure 6: Average mIoU *w.r.t.* total training cost for four semantic segmentation datasets. All models use CLIP as pre-trained weights.

### 4.2.2 MULTI-DATASET PANOPTIC SEGMENTATION

Considering the high training cost of panoptic segmentation, we only give a comparison of LMSeg with other multi-dataset training methods. As shown in Table 3, for the more challenging panoptic segmentation task, LMSeg outperforms "multi-head" and "manual-taxonomy" by a large margin in various settings.

## 4.3 ABLATION STUDY

**Weight Initialization.** LMSeg learns to align text embedding and segmented query embedding in the same embedding space. This subsection shows that Vision-Language pre-training (VLP) benefits LMSeg, as VLP helps initialize an aligned embedding space for vision and language. As shown in Table 4, when the text encoder is initialized with CLIP, and the image encoder is initialized with ImageNet pre-training, the experimental results correspond to language pre-training initialization.

| Pre-training weights | | Per-dataset mIoU | | | | Average mIoU |
|---|---|---|---|---|---|---|
| Text encoder | Image encoder | ADE20K | Cityscapes | COCO-Stuff | Mapillary Vistas | |
| CLIP-R50 | INT-R50 | 42.57 | 74.93 | 38.52 | 45.21 | 50.30 |
| CLIP-R50 | CLIP-R50 | 43.44 | 77.72 | 40.76 | 49.11 | 52.75 |

Table 4: Ablation experiments of VLP initialization in LMSeg. All models are trained with a batch size of 16 and a total number of iterations of 320k.

| Order of cross attention | Per-dataset mIoU | | | | Average mIoU |
|---|---|---|---|---|---|
| | ADE20K | Cityscapes | COCO-Stuff | Mapillary Vistas | |
| text-visual | 43.69 | 77.91 | 40.48 | 49.75 | 52.95 |
| visual-text | 44.37 | 78.20 | 40.53 | 50.57 | 53.41 |

Table 5: Experiments on cross-attention order in the CGD module. All models are trained with a batch size of 16 and a total number of iterations of 320k.

| CGD | DAA | Per-dataset mIoU | | | | Average mIoU |
|---|---|---|---|---|---|---|
| | | ADE20K | Cityscapes | COCO-Stuff | Mapillary Vistas | |
| | | 43.44 | 77.72 | 40.76 | 49.76 | 52.91 |
| ✓ | | 44.37 | 78.20 | 40.53 | 50.57 | 53.41 |
| ✓ | ✓ | 44.89 | 79.81 | 39.21 | 52.27 | 54.04 |

Table 6: Ablation experiments on proposed CGD and DAA. All models are trained on four semantic segmentation datasets with a batch size of 16 and a total number of iterations of 320k.

| CGD | DAA | Per-dataset PQ | | | Average PQ |
|---|---|---|---|---|---|
| | | ADE20K-Panoptic | Cityscapes-Panoptic | COCO-Panoptic | |
| | | 25.77 | 48.35 | 29.03 | 34.38 |
| ✓ | | 30.24 | 49.69 | 33.74 | 37.88 |
| ✓ | ✓ | 30.84 | 51.75 | 34.16 | 38.91 |

Table 7: Ablation experiments on proposed CGD and DAA. All models are trained on three panoptic datasets with a batch size of 16 and a total number of iterations of 320k.

When text and image encoders are both initialized with CLIP, the experimental results correspond to vision-language pre-training initialization. Experimental results show that LMSeg with VLP initialization significantly outperforms language pre-training initialization, improving the average mIoU on four datasets from 50.30 to 52.75.

**Category-guided Decoding Module.** There are two cross-attention layers in the category-guided decoding module. In this subsection, we study the effect of their order. "text-visual" indicates that the cross-attention module between segment query and text embedding is in front; otherwise "visual-text". As shown in Table 5, the experimental results show that "visual-text" performs better than "text-visual". "visual-text" is used by default.

**Factor-by-factor Ablation.** As shown in Table 6 and Table 7, the proposed category-guided decoding module increases the average mIoU on four datasets from 52.91 to 53.41 and the average PQ on three datasets from 34.38 to 37.88, verifying the module's effectiveness. The proposed dataset-aware augmentation strategy adopts separate suitable data augmentation for each dataset, thereby improving the model's overall performance. The average mIoU and average PQ increase from 53.41 to 54.04 and 37.88 to 38.91, respectively.

## 5 CONCLUSION

In this work, we propose a language-guided multi-dataset segmentation framework that supports both semantic and panoptic segmentation. The issue of inconsistent taxonomy is addressed by unified text-represented categories which not only reflect the implicit relation among classes but also are easily extended. To balance the cross-dataset predictions and per-dataset annotation, we introduce a category-guided decoding module to inject the semantic information of each dataset. Additionally, a dataset-aware augmentation strategy is adopted to allocate an approximate image preprocessing pipeline. We conduct experiments on four semantic and three panoptic segmentation datasets, and the proposed framework outperforms other single-dataset or multi-dataset methods. In the future, we will introduce more datasets and further study the zero-shot segmentation problem with complementary and shared categories across datasets.

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
