# OpenReview forum: "LMSeg: Language-guided Multi-dataset Segmentation"
_ICLR.cc/2023/Conference — ICLR 2023 poster_

### Official Review · Reviewer_wkpT · 2022-10-22

**Confidence:** 3
**Clarity, Quality, Novelty And Reproducibility:** The paper is well written. And I thin…
**Correctness:** 3
**Technical Novelty And Significance:** 2
**Empirical Novelty And Significance:** 2
**Recommendation:** 6

**Strength And Weaknesses:**

Pros:

1, The paper is well-written and easy to understand.
2, The proposed method is simple but efficient.
3, The experiments are sufficient.

Cons:

1, In the table2 of semantic segmentation, the proposed method seems to improve marginally if has a long training time. However, the table3 of panoptic segmentation improves largely.  Could the author have more analysis? is it related to the improvement of instance segmentation?

2, I note that the proposed method is benefited from both CLIP image encoder and text encoder. Thus I wonder if the main reason for the improvement of the proposed method is the usage of the CLIP model, especially the CLIP Image encoder.




**Summary Of The Paper:**

The paper proposes a language-guided framework for multi-dataset training. To solve the consistency of categories in each dataset, the paper encodes the categories in text embedding without manual reconciliation. Also, the paper adopts a dataset-aware augmentation strategy that assigns each dataset a specific image augmentation pipeline, which can suit the properties of images from different datasets.
The extensive experiments demonstrate that the proposed method achieves significant improvements on four semantic and three panoptic segmentation datasets.

**Summary Of The Review:**

The proposed method gives new insight, using a text encoder to solve the inconsistency of the label system in the different datasets, in multi-dataset segmentation training. This is an interesting direction. In my view, the more experiments analysis should be added.

---

> ### Author Response · Authors · 2022-11-07
> **For Reviewer wkpT**
>
> Thanks for these valuable comments.
>
> **Q1:** Why is the improvement of panoramic segmentation more obvious?
>
> **A1:**
> | Dataset | max number of segment objects per image |
> | --- | --- |
> | ADE2OK-semantic | 31 |
> | ADE20K-panoptic | 208 |
>
> As shown in the table above, the things categories in panoptic segmentation greatly increases the number of objects in the image that need to be predicted. In this case, the role of our proposed CGD module will be more obvious.
>
> Specifically, for things categories, a category may have multiple instances, which makes MaskFormer-based multi-dataset panoptic segmentation more difficult because the model has a limited number of output predictions (default 100 queries).
> The role of our proposed CGD module in the panoptic segmentation task becomes more pronounced, allowing the model to output segmentation results for the specified categories when the number of queries is limited.
>
> As shown in the following table, corresponding to table 7 in our paper, we further analyze the PQ metric of the things and stuff categories. The CGD module improves the PQ metric of both things and stuff categories, and the improvement of things categories is more prominent.
>
> | Method | ade panoptic | cityscape panoptic | coco panoptic |
> | --- | --- | --- | --- |
> | Ours (w/o CGD) | 25.77 (things: 22.00, stuff: 33.32) | 48.35 (things: 32.29, stuff: 60.03) | 29.03 (things: 29.37, stuff: 28.52) |
> | Ours (w CGD) | 30.24 (things: 27.77, stuff: 35.17) | 49.69 (things: 34.49, stuff: 60.73) | 33.74 (things: 34.49, sutff: 32.60) |
>
>
> There is a similar trend in Table 3 in our paper, as shown in the table below.
>
> | Method | ade panoptic | cityscape panoptic | coco panoptic |
> | --- | --- | --- | --- |
> | Manual-taxonomy | 30.68 (things: 28.24, stuff: 35.57) | 53.47 (things: 36.51, stuff: 65.81) | 36.63 (things: 38.55, stuff: 33.72) |
> | Mulit-head | 31.99 (things: 29.30, stuff: 37.36) | 54.17 (things: 38.68, stuff: 65.43) | 35.18 (things: 36.60, stuff: 33.04) |
> | Ours | 35.43 (things: 33.16, stuff: 39.96) | 54.77 (things: 39.34, stuff: 65.99) | 38.55 (things: 40.88, stuff: 35.04) |
>
>
> **Q2:** CLIP pretrain.
>
> **A2:**  All methods in Table 2 and Table 3 use CLIP to initialize the image encoder, so the comparison is fair.

---

### Official Review · Reviewer_DHCs · 2022-10-23

**Confidence:** 4
**Correctness:** 4
**Technical Novelty And Significance:** 2
**Empirical Novelty And Significance:** 2
**Recommendation:** 3

**Clarity, Quality, Novelty And Reproducibility:**

The illustration of this paper is clear and easy to understand. However, the originality of this paper is limited and the improvements are also not impressive to me. Although the authors attach their configs in the supplementary material, they do not promise to release the code in the paper so the reproducibility of this work cannot be guaranteed.

**Strength And Weaknesses:**

### Strength

1. The paper is well written and easy to follow.

2. I recognize the significance of the problem targeted in this paper. It should be feasible to identify instances in images of datasets labelled for semantic segmentation.

### Weaknesses

**1. The novelty is limited.** The main claim of the authors is that they propose *text-query alignment to address the issue of taxonomy inconsistency*. However, introducing language embeddings to build a unified label space for image segmentation has already been proven to be effective [1, 2, 3]. The Category-guided decoding (CGD) here is just another variant of LSeg [1] that exploits the text-pixel affinity to bridge the domain gap across different label spaces. Dataset-aware augmentation (DAA) here is a non-learnable strategy for data augmentation, which seems to be better mentioned in the section of implementation details instead of methodology.

**2. The improvement is not strong enough.** The numbers shown in Table 2 are not impressive from my point of view. For example, the gain for the average mIOU across four datasets can only be shrunk to a negligible 0.2% for 640K. I focus more on the results trained with a longer schedule since MaskFormer is hard to optimize and originally it needs 300 epochs to train on COCO panoptic. The results look to be prettier when trained under a short schedule, however, this might be due to a better initialization as the encoder of the proposed method is first pre-trained by CLIP (as discussed in Table 4).

### Reference

[1] Boyi Li, Kilian Q Weinberger, Serge Belongie, Vladlen Koltun, and Rene Ranftl. Language-driven semantic segmentation. ICLR 2022

[2] Xu, Mengde, et al. "A simple baseline for zero-shot semantic segmentation with pre-trained vision-language model." arXiv preprint arXiv:2112.14757 (2021).

[3] Rao, Yongming, et al. "Denseclip: Language-guided dense prediction with context-aware prompting." CVPR 2022.

**Summary Of The Paper:**

This paper presents LMSeg, which aims to train a model for image segmentation on multi-datasets. The authors claim that they resolve two major challenges in this paper:
- 1 the domain gap or inconsistency between semantic segmentation and panoptic segmentation.
- 2 the separated label space.

Experiments show that the proposed method can achieve better results than the previous baseline.

**Summary Of The Review:**

My concerns primarily lie in the novelty and efficacy of the proposed method. As discussed above, neither the story nor the performance of this paper impress me, I am thereby inclined to reject this paper.

---

> ### Author Response · Authors · 2022-11-07
> **For Reviewer DHCs**
>
> Thanks for these valuable comments.
>
> **Q1:** Difference to existing work using text embedding in segmentation.
>
> **A1:**
> We admit that the idea of using text embedding to segmentation tasks has been introduced previously, and we also have cited some work in our related work. However, here, we want the reviewer to understand our new contributions:
>
> 1) Text-Qurey alignment. Existing work extends CLIP's text-image contrastive to text-pixel contrastive learning for semantic segmentation. However, text-pixel contrastive is not suitable for instance or panoptic segmentation tasks. To this end, we firstly propose the text-query contrastive in a DETR-like segmentation framework, supporting both semantic and panoramic segmentation tasks.
>
> 2) Category-Guided Decoding (CGD) module. Taking the instance segmentation as an example, suppose we have an image with 100 persons and 100 bicycles. In the dataset of segmenting persons, the image has only person annotations, and the model's 100 predictions are required to segment out as many persons as possible. However, in the dataset of segmenting bicycles, the image only has bicycle annotations, and the model's predictions must be bicycles. When jointly training the two datasets, the model's predictions oscillate between persons and bicycles, impacting the model's performance on both datasets. We find that the root of the problem lies in decoupling the image branch and the text branch. To this end, we propose a simple yet effective category-guided decoding (CGD) module, which dynamically guides the model's predictions to the specific categories of each dataset. Experimental results on semantic segmentation (Table 6) and panoptic segmentation (Table 7) show that our proposed CGD can very effectively improve the average performance of the model on multi-dataset training.
>
> 3) Dataset-Aware Augmentation (DAA). We argue that DAA is an essential issue with significant effects on performance (as shown in Tables 6 and 7) but has always been ignored in previous multi-dataset training methods. Therefore, we add DAA to the METHOD section, emphasizing its importance in multi-dataset training.
>
> Furthermore, the reviewer may not really understand the role of our proposed CGD module, confusing it with text-query alignment that is ultimately used for classification.
>
>
> **Q2:** Question about the performance.
>
> **A2:** First, methods in Table 2 and Table 3 all use CLIP to initialize the image encoder, and all use dataset-aware augmentation, so the comparison is fair.
>
> Second, the improvement in Table 2 is not evident because the number of appeared categories per image is small. As shown in the table below, we count the ADE20K semantic dataset, and the maximum number of categories in one image in the training set is only 31. Even considering there may be unlabeled categories from other datasets, the default 100 queries of MaskFormer are also enough. Thus, the effect of our proposed CGD module is not obvious.
> However, in the multi-dataset panoptic segmentation task, the situation changes. As shown in the table below, even in the ADE20K panoptic dataset, up to 208 segmentation objects may appear in one image. The situation is more complicated when considering categories from other datasets.
> In multi-dataset panoptic segmentation, our proposed CGD module can play an obvious role. By knowing which categories to predict with text-query cross-attention in advance, we can make more efficient use of the limited number of queries of MaskFormer. The ablation experiments in Tables 6 and 7 can verify this.
>
> | Dataset | max number of segment objects per image |
> | --- | --- |
> | ADE2OK-semantic | 31 |
> | ADE20K-panoptic | 208 |
>
>
>
> **Q3:** Question about the code.
>
> **A3:** We promise to release the code after the work is published.

---

> > ### Comment · Reviewer_DHCs · 2022-11-22
> > **Thanks for your response**
> >
> > I appreciate the thorough and valuable response from the authors. My concern on the novelty issue of CGD is basically resolved. However, some of my concerns are still unresolved:
> >
> > **1. Regarding the novelty issue of text-query alignment.**
> >
> > I am not convinced by the answer that the text-query is different from the previous CLIP-based text-image segmentation framework. If I understand correctly, the query is a set of learnable parameters and the text inputs will be linearly transformed by the decoded query. I did not find the detailed illustration of the query decoder as mentioned in Figure 1(c). The description in Section 3.3 shows that the dot product between the query and the text embeddings is identical to a fully-connected layer. If we add an extra linear transformation on top of the text encoder of CLIP, the whole process of text modelling between [1, 2, 3] and this paper is almost the same.
> >
> > To my understanding, what differs the most between this paper and the previous CLIP-based papers is the contrastive loss after the dot product. However, there are already papers demonstrating the efficacy of contrastive loss in image segmentation [4, 5], which also weakens the novelty of this paper.
> >
> > **2. Regarding the performance.**
> >
> > The current results cannot well demonstrate the significance of multi-dataset segmentation. In Table 3, the author show that the proposed method can work well on panoptic segmentation. Accoridng to the original Maskformer paper, the PQ for COCO panoptic segmentation is 46.5 (+8 than the proposed method) when trained with COCO panoptic training set. I admit there should be domain gap among different datasets, but can we first start on datasets that are from the same source, e.g. training on COCO instance and COCO-stuff then evaluating on COCO panoptic? The huge gap between the proposed method and the baseline makes the results much less impressive.
> >
> > **3. Regarding the novelty issue of DAA.**
> >
> > I still inclined to move it to the implementation details but it's OK if the authors insist to regard it as a method.
> >
> > Hope the above comments can help further improve the manuscript.
> >
> > ### Reference
> >
> > [4] Wang W, Zhou T, Yu F, et al. Exploring cross-image pixel contrast for semantic segmentation, CVPR 2021
> >
> > [5] Wang X, Zhao K, Zhang R, et al. ContrastMask: Contrastive Learning to Segment Every Thing, CVPR 2022.

---

> > > ### Author Response · Authors · 2022-11-23
> > > **For Reviewer DHCs -- V2**
> > >
> > > Thank you very much for your patient and valuable reply.
> > >
> > > **Q1:** Regarding the novelty issue of text-query alignment.
> > >
> > > **A1:**
> > >
> > > 1). First of all, thank you for acknowledging the novelty of CGD module.
> > >
> > > 2). About Text-Query Alignment. We admit the relatively minor novelty of text-query alignment as a natural extension of text-pixel alignment. But this extension enables us to employ CLIP for multi-dataset semantic and panoptic segmentation.
> > >
> > >
> > >
> > >
> > > **Q2:** Regarding the COCO-panoptic performance
> > >
> > > **A2:**
> > >
> > > 1). Domain differences between different datasets are one reason, but more importantly, we did not employ a longer training setup. The experimental setups in Table 3 are able to verify the effectiveness of our multi-dataset training framework on the panoptic segmentation task.
> > >
> > > For panoptic segmentation tasks, Maskformer takes a long time to train on each dataset. For example, on the COCO-Panoptic dataset, MaskFormer was trained for 554k iterations with a batch size of 64 (with 64 GPUS), equivalent to 2216k iterations with a batch size of 16.
> > >
> > > In Table 3, we use three panoptic datasets for training. Our experimental settings are a batch size of 16 and a total number of iterations of 960k, where each dataset is allocated an average of 320k iterations. The 960k setting experiment took about four days on 8 A100 GPUs. Thus, the 960k is an appropriate experimental setting to effectively validate multi-dataset training frameworks for the panoptic segmentation task **without incurring huge training costs**.
> > >
> > > 2). Furthermore, MaskFormer requires less training time on semantic segmentation tasks. Experimental results in Table 1 demonstrate that our multi-dataset training framework can achieve non-inferior performance to single-dataset optimization.

---

### Official Review · Reviewer_G1t4 · 2022-10-31

**Confidence:** 4
**Clarity, Quality, Novelty And Reproducibility:** Good
**Correctness:** 3
**Technical Novelty And Significance:** 3
**Empirical Novelty And Significance:** 3
**Recommendation:** 5

**Strength And Weaknesses:**

Overall, this paper is good.

Strength:


1, The paper is well written and easy to follow.

2, The idea of using unified text embedding to enhance the object queries is interesting and novel for DETR-like framework for multiple datasets segmentation tasks. Category-guided decoding (CGD) is simple yet effective.

3, The extensive results show the effectiveness of proposed framework.


Weakness:

1, The idea of using text embedding for multi-dataset segmentation is not novel. The idea is proposed in the work “The devil is in the labels: Semantic segmentation from”(half year ago on arxiv. ) This paper adds learnable embedding (From COOP-IJCV-2022) and extra attention module. The author should not claim this is very huge contribution in introduction part.

2, Missing Experiment results M-Seg dataset. It would be better for benchmarking results on this dataset.

3, Missing ablation studies on CGD design.

4, Missing parameter and GFlops analysis on proposed CGD.


**Summary Of The Paper:**

This paper aims to solve two problems of multi-dataset segmentation: (1), inconsistent taxonomy. (2), inflexible of one-hot taxonomy. They map the taxonomy into embedding space with pre-trained pre-trained text encoder. Then they proposed a category-guided decoding module is designed to dynamically guide predictions to each dataset’s taxonomy.
Extensive experiments demonstrate that the proposed method achieves significant improvements on four semantic and three panoptic segmentation datasets


**Summary Of The Review:**

Overall, this paper is interesting to me. However, using text embedding for multi-dataset segmentation has been explored before.  Reminded by other reviewer opinions (Reviewer DHCs), I find the real technical novelty is very limited (Category-guided decoding (CGD) and Dataset-aware augmentation (DAA). I lower down my ratings.

---

> ### Author Response · Authors · 2022-11-07
> **For Reviewer G1t4**
>
> Thanks for these valuable comments.
>
> **Q1:** Difference to existing work using text embedding in segmentation.
>
> **A1:** We will highlight our new contributions in the introduction. We admit that the idea of using text embedding to segmentation tasks has been introduced previously, and we also have cited some work in our related work. Our new contributions are:
>
> 1) Text-Qurey alignment. Existing work extends CLIP's text-image contrastive to text-pixel contrastive learning for semantic segmentation. However, text-pixel contrastive is not suitable for instance or panoptic segmentation tasks. To this end, we firstly propose the text-query contrastive in a DETR-like segmentation framework, supporting both semantic and panoramic segmentation tasks.
>
> 2) Category-Guided Decoding (CGD) module. Taking the instance segmentation as an example, suppose we have an image with 100 persons and 100 bicycles. In the dataset of segmenting persons, the image has only person annotations, and the model's 100 predictions are required to segment out as many persons as possible. However, in the dataset of segmenting bicycles, the image only has bicycle annotations, and the model's predictions should be bicycles. When jointly training the two datasets, the model's predictions oscillate between persons and bicycles, impacting the model's performance on both datasets. We find that the root of the problem lies in decoupling the image branch and the text branch. To this end, we propose a simple yet effective category-guided decoding (CGD) module, which dynamically guides the model's predictions to the specific categories of each dataset. Experimental results on semantic segmentation (Table 6) and panoptic segmentation (Table 7) show that our proposed CGD can very effectively improve the average performance of the model on multi-dataset training.
>
>
> **Q2:** Missing Experiment results M-Seg dataset.
>
> **A2:** The MSEG dataset consists of 7 datasets, which is over 200 GB. To reduce the experimental burden, we selected four commonly used semantic segmentation datasets and three commonly used panoptic segmentation datasets. The experiments on these datasets can already verify the effectiveness of our proposed method.
> On the other hand, the advantages of our proposed CGD module should be more pronounced when the number of datasets is large. We will perform this time-consuming verification work in follow-up work.
>
>
> **Q3:** Missing ablation studies on CGD design.
>
> **A3:** We conduct related ablation experiments on the CGD module. Table 5 conducts the experiments on cross-attention order in the CGD module. Tables 6 and 7 conduct factor-by-factor ablation experiments for semantic and panoptic segmentation, respectively, both including the CGD module.
>
>
> **Q4:** Missing parameter and GFlops analysis on proposed CGD.
>
> **A4:** As shown in the table below, since CGD adds another text-query cross attention to the original decoder module, the number of parameters hardly increases, while the GFlops increase from 0.694 to 0.821.
> We will add the description of parameters and GFflops in the revised manuscript.
>
> | model | decoder parameters | decoder GFlops |
> | --- | --- | --- |
> | w/o CGD | 6.33M   |  0.694G |
> | w CGD | 6.33M    |    0.821G |

---

### Official Review · Reviewer_6V2p · 2022-11-11

**Confidence:** 5
**Correctness:** 4
**Technical Novelty And Significance:** 3
**Empirical Novelty And Significance:** 3
**Recommendation:** 6

**Clarity, Quality, Novelty And Reproducibility:**

The paper is very well written and easy to follow. With given experimental settings it should be simple to reproduce the listed evaluation.

**Strength And Weaknesses:**

Strengths:
1. Multi-dataset training for segmentation using a unified taxonomy is the first step towards open-set model. LMSeg automates this training without any -- additional parameters in the form of dataset-specific head, manual mapping and drop in per-dataset mIoU. Similar to other open-set vision model, LMSeg leverages language embeddings to build a unified model.
2. Moreover, the proposed LMSeg is simple, elegant and easy to implement straight out of MaskFormer model.

Weaknesses:
1. The model is well engineered. However, I feel there is limited novelty. Multiple recent work has resorted to text-query alignment to build an open-set model for different problem settings (object detection, referring expression). Motivated by this development, LMSeg modifies and trains Maskformer like framework with similar losses. To me, the only novel contribution seems category-guided decoder design.
2. Secondly, Table 2 & 6 suggests that the impact of LMSeg, CGD & DAA framework on semantic segmentation setting is very minimal. Hence the novel components only contributes to improvements in panoptic segmentation. Any thoughts on what causes this discrepancy ?

How about evaluating LMSeg in an open-world setting ? Or atleast, cross-category evaluation (i.e., train model by leaving out test categories) or datasets relevant to training data (such as ADE-Full) ? These results should strengthen the empirical evaluation. As for now the multi-dataset evaluation has very limited use case.

**Summary Of The Paper:**

This work extends a MaskFormer model to build LMSeg model, that learns to segment categories across multiple datasets using unified training without resorting to manual taxonomy. LMSeg is built on the fact that the text embeddings for semantically similar categories, as retrieved from CLIP text encoder, projects close to each other on euclidean space. Hence by learning to align the segment embeddings to the text embeddings, it is possible to learn to pool similar categories from multiple datasets without manual taxonomy mapping. Further, in order to dynamically guide the segment embeddings towards dataset-specific taxonomy, LMSeg proposes to use category-guided text-query cross-attention within the transformer decoder. This together with the dataset-aware augmentation strategy can train SoA multi-dataset model that outbeats single-dataset trained models (with similar total budget).

**Summary Of The Review:**

LMSeg work makes a good read with clean experimental evaluation with few insights drawn during ablation. However, due to limited technical novelty I would place current version at Borderline acceptance. Adding extra evaluation and drawing better insights, would clearly add more value.

---

> ### Author Response · Authors · 2022-11-17
> **Refer to Reviewer-6V2p**
>
> Thanks for these valuable comments.
>
> **Q1:** Multiple recent work has resorted to text-query alignment to build an open-set model for different problem settings (object detection, referring expression).
>
> **A1:** About the text-query alignment.
>
> Although there are similar text-query alignment ideas in other tasks, they have not yet appeared in the segmentation task. As far as I know, CLIP is presently mainly used for semantic segmentation tasks through text-pixel alignment. However, compared with semantic segmentation, the panoptic segmentation task is more general.
>
> Although simple, our proposed text-query alignment is still instructive for panoptic segmentation tasks. Since for semantic segmentation tasks, we even don't have to use the proposed text-query alignment, but directly employ the text embeddings as object queries, which we have verified to be effective in semantic segmentation tasks.
>
>
> **Q2:** Hence the novel components only contributes to improvements in panoptic segmentation. Any thoughts on what causes this discrepancy ?
>
> **A2:**
>
> **About Dataset-Aware Augmentation (DAA)**.
> DAA is practical in both semantic and panoptic segmentation tasks, as shown by the ablation experiments in Tables 6 and 7. However, in Tables 2 and 3, our method and other methods all adopt DAA for a fair comparison.
>
> **About Category-Guided Decoder (CGD)**.
>
> The CGD module is more effective in panoptic segmentation task because:
>
> 1). In the panoptic segmentation task, the number of objects (stuff + things) to be segmented is much larger than that of semantic segmentation (stuff only).
>
> As shown in the table below, we count the ADE20K semantic dataset, and the maximum number of categories in one image in the training set is only 31. However, in the multi-dataset panoptic segmentation task, the situation changes. As shown in the table below, even in the ADE20K panoptic dataset, up to 208 segmentation objects may appear in one image.
>
> | Dataset | max number of segment objects per image |
> | --- | --- |
> | ADE20K-semantic | 31 |
> | ADE20K-panoptic | 208 |
>
> 2). Our proposed CGD module allows the model's predictions to focus on the classes that need to be predicted, making more efficient use of limited object queries (100 by default).
>
> In this experiment, we use the same model and input images and observe the difference in model predictions by varying the input text embeddings in the CGD module. Note that the final classification of object queries is still a (K+1)-way classification, where the number of categories K is 133 for coco-panoptic. **We only change the input text embeddings used in the CGD module for cross-attention with object queries**, while the text embeddings used later for classification remain unchanged.
>
> The image is selected from the coco-panoptic validation set (000000184324.png).
>
>
> If we don't change the text embedding input to CGD, the model's top 10 predictions are：
> ```
> top10_scores: [0.99, 0.98, 0.98, 0.98, 0.97, 0.96, 0.95, 0.95, 0.95, 0.93]
> top10_labels: [119,0,0,0,0,0,0,129,0,0]
>
> where classes 0, 119 and 129 correspond to person, sky-other-merged and building-other-merged, respectively;
> ```
>
> Assume that the input text embeddings of the CGD module contain only person and background. In this case, the model's top 10 predictions change as follows, with predictions for other classes suppressed:
> ```
> top-10 scores: [0.89, 0.84, 0.80, 0.75, 0.70, 0.67, 0.66, 0.65, 0.64,
>         0.63]
> top-10 labels: [0,0,0,0,0,0,0,0,0,0]
> ```
>
> This experiment demonstrates that our CGD module can dynamically adjust the model's outputs to given categories.
>
> **Q3:**
> How about evaluating LMSeg in an open-world setting ?
>
> **A3:** Zero shot performance.
>
> We add the following experiments, using Cityscapes + COCO-stuff  + Mapillary-vistas as the training set and ADE20K as the test set.
>
> We train a model configured for 320k iteration training, and the results on ADE20K test set are shown in the following table:
>
> | mIoU of 52 appeared-classes | mIoU of 98 non-appeared-classes |
> | --- | --- |
> | 32.14 | 6.18 |
>
> (The model's training is not over yet, and the above is the intermediate result when iterating 154k.)
>
> We divide the 150 categories of ADE20K into two types: the 52 categories that have appeared in the training set and the 98 categories that have never appeared in the training set. The mIoU of the 98 non-appeared-classes is much lower than the 52 appeared-classes.
>
> This result is reasonable because a **category-agnostic RPN module** is first trained in the zero-shot object detection framework to provide box proposals for all categories of objects. Then CLIP is utilized to classify these boxes in a zero-shot style. A contemporary work [1] focusing on open-world panoptic segmentation also first trains a class-agnostic mask Proposal network. However, our framework is designed for multi-dataset segmentation, and the object queries are not designed to be **category-agnostic**.
>
> [1] Open-Vocabulary Panoptic Segmentation with MaskCLIP

---

### Decision · Program_Chairs · 2023-01-20

**Decision:**

Accept: poster

**Justification For Why Not Higher Score:**

The novelty is indeed limited, and the application is a bit niche at the moment.

**Justification For Why Not Lower Score:**

There is enough novelty/interesting findings here to justify acceptance.

**Metareview: Summary, Strengths And Weaknesses:**

The reviewers are nominally split, so the average rating is borderline. But, reading the reviews, it seems clear that the main shared concern is the novelty. More specifically, as the authors acknowledge (both in the paper and in the discussion) the basic idea of using language embeddings for segmentation targets is of course not new; but there is also a clearly novel component here, namely the CGD (which allows multi-dataset application). So the question is how significant is this novelty. The reviewers are somewhat lukewarm about it, but acknowldedge it as a novel contribution. Furthermore, reviewer DHCs, who gave the lowest score, with limited novelty cited as the chief concertn, does agree after the discussion that CGD is novel.

So the final judgment seems to come down to the significance of the results (how "impressive" they are). There is a detailed discussion of these, with the authors explaining the fairness of comparison and the relevance of improvements observed with short training time. I agree that these results are meaningful. The main contribution of this paper is clearly not a new "SOTA" number but a new technical idea for dealing with multiple datasets.

I have read the paper and the reviews,rebuttals, and responses. I believe this submission is clearly above bar; as the field moves from fixed/arbitrary datasets to dealing with open-world data via data streams with images (and other data) from mixed origins, it will become increasingly important to explore tools like the system proposed here.



**Note From Pc:**

if the above contains the word "oral" or "spotlight" please see: "oral" presentation means -> notable-top-5% and "spotlight" means -> notable-top-25%. As stated in our emails, we are disassociating presentation type from AC recommendations